# Associations between Hunger and Psychological Outcomes: A Large-Scale Ecological Momentary Assessment Study

**DOI:** 10.3390/nu14235167

**Published:** 2022-12-05

**Authors:** Romain de Rivaz, Joel Swendsen, Sylvie Berthoz, Mathilde Husky, Kathleen Merikangas, Pedro Marques-Vidal

**Affiliations:** 1Department of Medicine, Internal Medicine, Lausanne University Hospital and University of Lausanne, 46 Rue du Bugnon, 1011 Lausanne, Switzerland; 2University of Bordeaux INCIA, EPHE PSL Research University, 146 Rue Léo Saignat, F-33000 Bordeaux, France; 3University of Bordeaux INCIA CNRS UMR 5287, 146 Rue Léo Saignat, F-33000 Bordeaux, France; 4Department of Psychiatry for Adolescents and Young Adults, Institut Mutualiste Montsouris, 42 Boulevard Jourdan, F-75014 Paris, France; 5Laboratoire de Psychologie EA4139, University of Bordeaux, 3 Ter Place de la Victoire, 33000 Bordeaux, France; 6Genetic Epidemiology Research Branch, National Institute of Mental Health, 35A Convent Drive, MSC #3720, Bethesda, MD 20892-9663, USA

**Keywords:** Ecological Momentary Assessment, hunger, food intake, psychology, epidemiology

## Abstract

Studies assessing the association between hunger and psychological states have been conducted in laboratory settings, or limited to persons with eating disorders. In this study, 748 community-dwelling adults (56.4% women, 60.0 ± 9.3 years) completed the Ecological Momentary Assessment four times a day (08:00, 12:00, 16:00 and 20:00) for seven days. At each assessment, respondents indicated their current hunger level, food intake, and psychological states (sad, anxious, active, lively, distracted, anhedonic, angry, slow thinking and restless). Time-lagged associations assessing the effect of hunger on subsequent psychological states 4 h later and vice-versa were assessed. Hunger intensity increased subsequent active feeling (coefficient and 95% confidence interval: 0.029 (0.007; 0.051)) and lively feeling (0.019 (0.004; 0.034)) and decreased later slow thinking (−0.016 (−0.029; −0.003)). Previous eating increased later activity (0.116 (0.025; 0.208)). Feeling active (0.050 (0.036; 0.064)), lively (0.045 (0.023; 0.067)) and restless (0.040 (0.018; 0.063)) increased later hunger intensity, while distraction (−0.039 (−0.058; −0.019)) and slow thinking (−0.057 (−0.080; −0.034)) decreased it. No association was found between hunger, food intake and negative psychological states (sadness, anxiety and anger). Conclusions: Positive psychological states and hunger influence each other, while no association was found between hunger and negative psychological states.

## 1. Introduction

A number of theories have been proposed to explain how the physiological state of hunger is linked to psychological constructs. Whereas the majority of studies to date have focused on the role of emotions or perceived stress as direct predictors of eating behavior (for meta-analyses, see [1]), there has been limited attention to the specific influence of hunger on psychological states. Hunger may increase aggressive feelings [2], risk taking [3], and interoceptive awareness [4], as well as decrease prosocial behavior or attitudes toward others [5]. The most frequently-cited reason for these changes is fluctuations in blood glucose levels [6,7,8,9] that may impact psychological states because hunger itself can be perceived as a negative emotion (i.e., feeling “hangry”) depending on the context [10]. Hunger can easily deplete emotional or psychological resources needed for self-control [11], and it may also influence emotions as a function of ghrelin levels [12] or genetic background [13].

Over the past two decades, the Ecological Momentary Assessment (EMA) has emerged as a powerful alternative for investigating psychological states through the collection of data in real time and in the natural contexts of daily life. EMA largely overcomes retrospective recall biases and allows researchers to examine the direction of associations among correlated variables. A recent review of EMA studies indicates strong overall support for emotion regulation models, whereby eating disorder behaviors temporarily reduce negative affect or increase positive affect [14]. Still, very few studies using EMA have specifically focused on hunger and its association with psychological states, and the majority of these studies have been limited to individuals with eating disorders (for a review, see [15]).

Further to the need of better understanding the association between hunger and psychological states in the general population, such information should be interpreted relative to the natural daily fluctuations of these variables. The few existing EMA studies of hunger in healthy individuals have shown important within-day variation, with expected peaks at lunch and dinner time [16], but also with a stronger association with perceived stress in the late afternoon or evening than earlier in the day [17]. Further, the assessment of associations between hunger and psychological states should also account for within-day fluctuations in psychological states.

The possibility that psychological states may be linked with hunger in population-based, community-dwelling samples has relevance for understanding the potential etiologic mechanisms involved in the transition from “normal” emotion-influenced eating to disordered eating behaviors that involve emotion-based processes. A better understanding of these associations could help to manage people with eating disorders and/or obesity.

Therefore, the objective of this investigation was to characterize the within-day associations between hunger and psychological states in a general population sample, adjusted for clinical covariates and recent eating episodes, and taking into account within-day fluctuations.

## 2. Materials and Methods

### 2.1. Participants

Participants were drawn from the CoLaus|PsyCoLaus study, a prospective study conducted in a representative, non-stratified sample of the population aged 35–75 years living in Lausanne, Switzerland. The objectives and details of the CoLaus|PsyCoLaus study have been published previously [18,19] and can be consulted at www.colaus-psycolaus.ch. During the second follow-up of the cohort (from May 2014 to April 2017), all participants contacted after first of June 2015 (start of the EMA study) were considered as eligible. Participants were excluded if they had less than 70% of the EMA data available, or if they failed to complete assessments for any covariate examined in the present study.

### 2.2. Ecological Momentary Assessment

The EMA methods and questions were based on items used in the National Institute of Mental Health (NIMH) Family Study of Affective Spectrum Disorders [20]. A Samsung Galaxy smartphone was dedicated to the study and used to obtain real-time assessments of psychological variables, the experience of hunger, and eating behavior. Participants were asked to respond to the EMA for seven consecutive days, and four fixed assessments were provided each day at 08:00, 12:00, 16:00 and 20:00. Fixed assessment times were used for each participant with an average delay of 4 h between assessments. Participants were allowed to answer questions within a 30 min window following each EMA signal. Based on the mood circumplex model [21], 7-point bipolar scales were used at each EMA assessment to assess psychological states ranging from (1) very happy (or relaxed, inactive, tired, focused, enjoying experiences), to (7) very sad (or anxious, active, lively, distracted, experiencing no pleasure). While debate persists concerning the assessment of mood through unipolar versus bipolar scales, the mood theory used in this investigation conceptualizes psychological states as existing on dimensions that can be assessed in this bipolar format. Daily event negativity assessment was based on the inventory of small life events that was adapted for use in EMA [22,23]. Participants were first asked to describe the one event that had the greatest impact on themselves since the previous assessment, then to rate it using a similar scale of event impact ranging from (1) extremely positive to (7) extremely negative. In addition to bipolar scales, two unipolar ones were used to assess anger and restlessness from (1) not at all angry/irritable (or restless) to (7) very angry/irritable (or restless). Based on these scales, participants were therefore able to report a wide range of psychological states that were negatively valenced (sad, anxious, tired, angry, restless, distracted, lack of pleasure) or positively valenced (happy, relaxed, lively, focused, pleasure). At each assessment, the participants were also asked to rate their experience of hunger at that moment from (1) not at all hungry to (7) extremely hungry, and to indicate if they had eaten anything since the last assessment (i.e., previous eating). If participants endorsed food consumption since the last assessment, they were asked to indicate the largest quantity of food consumed during this period (snack, small meal, medium meal, or large meal). The list of questions related to the mood circumplex (in French) and the corresponding questions in the NIH family study are provided in Appendix B.

### 2.3. Other Covariates

Covariates were collected during the same examination when participants were invited to participate in the EMA study. Participants reported their age, sex, marital status, current medications, and smoking and alcohol consumption by questionnaire. Marital status was categorized into married or other (i.e., single, divorced, widowed); smoking status was categorized into never, former (irrespective of the time since quitting) and current (irrespective of the amount smoked). Alcohol consumption was categorized into consumer and teetotaler.

Weight and height were measured with participants barefoot and in light clothes. Body weight was measured in kilograms to the nearest 100 g using a Seca^®^ scale (Hamburg, Germany). Height was measured to the nearest 5 mm using a Seca^®^ (Hamburg, Germany) height gauge. Due to the small number of participants with a body mass index (BMI) <18.5 kg/m^2^ (*n* = 12) they were included in the normal weight group. Overweight was considered if participant BMI was between 25 and 29.9 kg/m^2^; obesity was considered if the BMI was ≥30 kg/m^2^.

Diabetes was defined by a fasting plasma glucose ≥ 7 mmol (126 mg/dL) and/or presence of an antidiabetic drug treatment. Diabetes was considered as an important covariate as most associations between hunger and psychological states have been suggested to be due to inadequate glucose levels and might be exacerbated among subjects with diabetes [8,24].

Depressive symptomatology was assessed using the Centre for Epidemiologic Studies Depression scale (CES-D), a 20-item questionnaire covering depressive symptoms experienced over the past week. A cutoff score ≥ 17 was used to estimate the presence or absence of depression. This value provides a sensitivity of 76% and a specificity of 71% in men [25]. In a separate interview, participants were queried regarding the presence of eating disorders (anorexia or bulimia) and if their condition was still active or remitted.

### 2.4. Statistical Analysis

Statistical analyses were performed using the Stata version 16.1 for windows (Stata Corp, College Station, TX, USA). Descriptive results were expressed as number of participants (percentage) for categorical variables and as average ± standard deviation or median (interquartile range) for continuous variables. Between-group comparisons were performed using chi-square or Fisher’s exact test for categorical variables and Student’s *t*-test or Kruskal-Wallis test for continuous variables.

The levels of hunger and psychological state for each assessment period (08:00, 12:00, 16:00 and 20:00) were computed using a mixed model taking into account repeated measures for each participant. Briefly, the following equation was considered:outcome_ij_ = β_0_ + β_i_ period_ij_ + u_j_ + ε_ij_
for i = 1,…,4 assessment periods and j = 1,…,*n* number of participants, with u_j_ = random effect for participant ID_j_. Where variable ID corresponds to the participant’s identification and categorical variable period corresponds to the four assessment periods of interest. Results were expressed as average ±standard error.

The bidirectional associations between hunger and psychological states and vice-versa were assessed as follows: first, the association between each psychological state at time T (dependent variable) and the hunger score at time T − 1 (independent variable) was modelled using a repeated-measures analysis, taking into account the participant and the assessment period, using a multilevel mixed model. We considered a random intercept and coefficient for the association between each psychological state and the lagged hunger score within each participant, and regarded the effect of assessment period as nested within each participant. We also considered the covariance matrix for the random part of the model to have distinct variances and covariances for each random effect. Three models were applied, taking into account the participant and assessment period, and adjusting for confounders. Model 1 adjusted for sex (male/female), age (continuous), marital status (married/other), smoking (never/former/current), BMI categories (normal/overweight/obese), diabetes (yes/no), alcohol consumption (yes/no), depressive status (yes/no) and previous eating as reported in EMA (yes/no). Model 2 included all confounders of Model 1 plus assessment period (12:00, 16:00 or 20:00). Model 3 included all confounders of Model 2 plus the psychological state assessed at time T − 1. This model was created to address the possibility that previous psychological states could influence the current state, and therefore erroneously influence conclusions of directionality. Model 4 included all confounders of Model 3 plus quantity of food consumed. For models 1 to 4, the results were expressed as a coefficient with 95% confidence intervals (CIs).

The association between the hunger score at time T and the different psychological states at time T − 1 was modelled similarly, except that the hunger score was considered as the dependent variable and each psychological state as the independent variable. The results were expressed as coefficients with 95% CIs. Significant results were considered for a two-sided test with *p* < 0.05.

### 2.5. Ethical Statement

The institutional Ethics Committee of the University of Lausanne, which afterwards became the Ethics Commission of Canton Vaud (www.cer-vd.ch) approved the baseline CoLaus study (reference 16/03). The approval was renewed for the first (reference 33/09), the second (reference 26/14) and the third (reference PB_2018-00040) follow-ups. The approval for the entire CoLaus|PsyCoLaus study was confirmed in 2021 (reference PB_2018-00038, 239/09). The study was performed in agreement with the Helsinki declaration and its former amendments, and in accordance with the applicable Swiss legislation. All participants gave their signed informed consent before entering the study.

## 3. Results

### 3.1. Selection Procedures and Characteristics of Participants

The selection procedures are summarized in Figure 1. Of the initial 4881 subjects who participated in the second follow-up of the CoLaus study, 1225 (25.1%) were recruited before the beginning of the EMA study and considered as non-eligible. Of the remaining 3656 eligible participants, 2690 (73.6%) declined, 145 (4.0%) were excluded because less than 70% of the EMA data were available, and 73 (2.0%) were excluded because of missing covariates. The final sample consisted of 748 participants, representing 15.3% of the total sample and 20.4% of the sample eligible to participate in the EMA study.

The characteristics of the participants included and excluded are summarized in Table 1. Included participants were younger, more frequently married, had a lower rate of diabetes and were more frequently alcohol consumers than excluded participants. Only two participants (0.3% of the final sample) reported remitted bulimia, and none reported anorexia. Hence, this information was not considered in the statistical models.

### 3.2. Levels of Hunger and Psychological States According to Assessment Period

The levels of hunger and psychological states according to the time of day are summarized in Appendix A. Participants reported being most hungry at 12:00 and the least hungry at 16:00; the hunger level was also low at 20:00. One quarter of participants reported being in the process of eating when the evaluation occurred at 12:00 and 20:00 (Appendix A). At 08:00 and 12:00, two out of three participants responded having eaten nothing or only a small snack since the last evaluation, while at 16:00 and 20:00, almost half responded that they had since eaten a normal or a large meal (Appendix A).

The intensity of sadness varied slightly according to the time of day, with the highest value occurring at 08:00. Participants were least anxious at 20:00, after a decreasing pattern throughout the day (with similar values for 12:00 and 16:00). Feeling active showed an inverse U curve, being highest at 12:00 and 16:00 and lowest in the morning (08:00) and the evening (20:00). A similar pattern was found for feeling lively and feeling distracted, although variations were smaller between the daytime peak and the morning/evening lows. Anger was stable throughout the day, with a slightly lower value at 20:00. Slow thinking showed a U curve, with higher values in the morning and the evening. Anhedonia was highest at 08:00 and lowest at 20:00, with a plateau between 12:00 and 16:00. Finally, restlessness was highest at 12:00 and lowest at 20:00.

### 3.3. Hunger Influencing Later Psychological States

The association of hunger level and eating at time T − 1 with psychological states at time T are summarized in Table 2. In the model 1 (adjusting for sex, age, marital status, smoking, BMI categories, diabetes, alcohol consumption, depressive status and previous eating), hunger intensity was positively associated with sadness, anxiety, feeling active and lively, and negatively associated with feeling distracted and slow thinking. After further adjustment for assessment period and previous psychological state, only the positive associations between hunger intensity and feeling active and lively, and the negative association between hunger and slow thinking, remained significant. Those findings were confirmed after further adjustment for previous quantity of food consumed.

Previous eating was positively associated with feeling active and lively, and negatively associated with feeling distracted, anhedonia and slow thinking. After further adjustment for assessment period and previous psychological state, only the positive association between previous eating and feeling active remained statistically significant. Finally, after further adjustment for quantity of food consumed, only the negative associations between previous eating and anxiety and anhedonia were statistically significant.

### 3.4. Psychological States Influencing Later Hunger

The effects of psychological states at time T − 1 on the hunger score at time T are summarized in Table 3. Feeling active or lively, anhedonia and restlessness were positively associated with later hunger intensity, while feeling distracted and slow thinking tended to be negatively associated with later hunger. After further adjustment by assessment period and previous mood, the positive associations between feeling active or lively and restlessness with later hunger intensity remained significant, and the negative associations between feeling distracted and slow thinking with hunger intensity increased in magnitude. Those associations were confirmed after further adjustment for quantity of food consumed.

## 4. Discussion

After multivariate adjustment, our results indicate that hunger is associated with changes in psychological states toward active (vs. inactive) and lively (vs. tired) psychological states, while eating was associated with shifts toward active states only. Our results also indicate that several psychological states influence hunger: feeling active, lively or restless increased hunger, while feeling distracted and slow thinking reduced hunger in the following hours.

### 4.1. Levels of Hunger and Psychological States According to Assessment Period

A prerequisite to understanding the covariation of hunger with psychological states is information concerning their natural variation as distinct variables. Hunger intensity peaked at midday and reached a nadir in the evening. This distribution is likely to be related to the timing of the main meals of the day, as responses at midday occurred when most participants had consumed little if any food, while the responses at night occurred when most participants had eaten a sizable amount. Psychological states also varied according to time of day, but with differing patterns. Feeling active, lively and distracted showed an inversed U-shaped curve, and the opposite pattern was observed for slow thinking. These findings are in agreement with the literature [26,27], as healthy subjects have been shown to display an inverted U-shape curve for positive and a U-shape curve for negative feelings, although the peak for negative feelings occurred earlier than in our study [26]. Conversely, sadness/happiness and anger varied little during the day.

### 4.2. Hunger Influencing Later Psychological States

Greater hunger intensity was prospectively associated with increasing active and lively states, and with decreases in slow thinking over the course of the day. These findings are in partial agreement with the concept that hunger increases ghrelin and leads to a more aggressive, “active” behavior [10,12]. Conversely, no association was found between hunger intensity and later shifts towards negative psychological states such as sadness or anger. Our findings contrast with previous studies [4,10] in which hunger led to negative emotions and judgements. A possible explanation for this discrepancy is that our participants likely experienced less intense hunger than those reported in other studies. For instance, the participants of one study had to fast for a whole day [4], while those from another study were made subject to 8 days of intense caloric restriction [13], conditions that deviate from normal daily activities. Overall, our results suggest that in a free-living, community-based sample, hunger may lead towards positive psychological states.

Previous eating was positively associated with shifts towards feeling active. This finding confirms past observations regarding a decrease in fatigue levels after snack consumption [28] and an increase in fatigue after skipping breakfast [29]. A possible mechanism is that subjects with olfactory and gustatory deficits are more prone to develop depression [30] or obesity [31].

However, after adjusting for quantity of food consumed, this association was no longer significant and the negative associations between previous eating and anxiety and anhedonia became significant. It should be stated that the significance levels of the associations were between 0.05 and 0.01, and would have not been considered as significant, had a correction for multiple comparisons been applied. Hence, our results suggest that the impact of dietary intake on subsequent psychological states is modest and the inconsistencies should be further investigated. In this respect, the potential impact that meal timing, sleep and their interaction have on psychological states should be clarified.

### 4.3. Psychological States Influencing Later Hunger

Feeling active or lively and restlessness were associated with greater hunger intensity at the next assessment period, while feeling distracted and slow thinking were linked to lower hunger intensity. While these findings partially replicate those of an interventional study where distraction decreased dietary intake [32], an opposite effect for distraction has also been reported [33]. However, these different findings are not necessarily contradictory in that distraction and slow thinking might deviate attention from food-related cues, thus decreasing hunger. It is also possible that, by increasing dietary intake, distraction will also lead to reduced hunger. Interestingly, little if no association was found between negative psychological states and later hunger intensity. These findings contrast with a meta-analysis indicating that negative psychological states increased hunger and food intake [1]. It is therefore important to note that most studies reported in this meta-analysis [1] were intervention studies, where negative psychological states were induced, whereas our study relied on a community-based sample assessed in naturalistic contexts. Furthermore, the previously observed effect of negative psychological states occurred mostly among restrained eaters, while it was not significant (contrary to positive psychological states) among non-restrained eaters [1]. Overall, our results suggest that in a free-living, community-based sample, shifts toward positive but not negative psychological states may induce hunger.

### 4.4. Strengths and Limitations

This is the largest study to date to use EMA in the assessment of hunger and psychological states in a community-dwelling sample. The large sample size and the multiplicity of measurements allowed identifying small but significant associations between hunger and psychological states. Further, the collection of data at different time points allowed taking into account the within-day fluctuations of these variables.

This study also has limitations. First, the data was self-reported and used scales. As a result, floor or ceiling effects may occur. Second, the sample represents only a subset of the participants in the Colaus|PsyCoLaus study. Hence, results might not be generalizable, and it would be important that our study be replicated. Third, we did not take into account environmental conditions such as location or physical activity, which have been suggested to affect psychological states [34,35]. Still, as participants provided information in a variety of locations and settings, the effect of environment is reduced. Fourth, EMA assessments occurred every four hours, and therefore effect of previous eating on hunger can be very different if it occurred almost four hours earlier or just before hunger was assessed. Our results should therefore be interpreted as the average effect over a four-hour period. Moreover, they do not take into account the potential interaction between food timing and sleep on psychological states [29,30]. Indeed, it has been shown that skipping breakfast is associated with increased fatigue [29]. Fifth, the CES-D has been developed to screen people with a high likelihood of depression in the general population in order to apply a diagnostic interview in a second step. Hence, we used this scale to classify participants depending on their risk of depression although this scale should normally not be used for diagnostic purposes. Finally, we did not correct for multiple analyses. Hence, it is possible that the inconsistent associations observed between previous eating at time T − 1 and psychological states at time T were due to chance, as the significance levels were all above 0.0125, which would be the Bonferroni threshold for four models (0.05/4)

## 5. Conclusions

By providing a test of the dynamic associations between psychological states, hunger and eating in a large community-based sample, the present study showed that positive psychological states and hunger influence each other, while no consistent association was found between hunger and negative psychological states. These findings obtained in people’s natural environment, using ambulatory monitoring, contribute to our understanding of proximal factors associated with hunger and eating behaviors. From a clinical perspective, examining whether different patterns of dynamic associations between affect, hunger, and eating are observed in people suffering from disordered eating behaviors, could be of importance for these patients’ care procedures.

## Figures and Tables

**Figure 1 nutrients-14-05167-f001:**
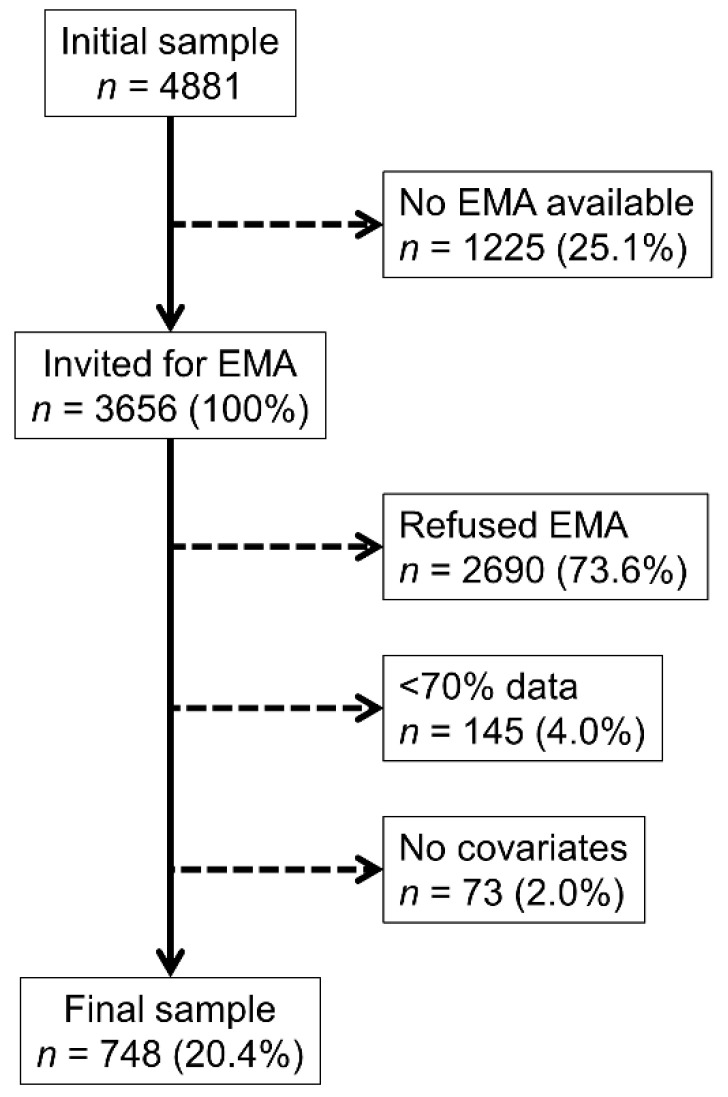
Selection procedure, CoLaus|PsyCoLaus study, Lausanne, Switzerland, 2015–2017.

**Table 1 nutrients-14-05167-t001:** Characteristics of included and excluded participants, CoLaus|PsyCoLaus study, Lausanne, Switzerland, 2015–2017.

	Included	Declined/Insufficient Data	*p*-Value
n	748	2908	
Female (%)	422 (56.4)	1595 (54.8)	0.442
Age (years)	60.0 ± 9.3	63.2 ± 10.4	<0.001
Age groups (%)			<0.001
45–54.9	261 (34.9)	785 (27.0)	
55–64.9	258 (34.5)	894 (30.7)	
65+	229 (30.6)	1229 (42.3)	
Married (%)	434 (58.0)	1475 (51.1)	0.001
Smoking status (%)			0.052
Never	318 (42.5)	1065 (40.9)	
Former	306 (40.9)	1004 (38.5)	
Current	124 (16.6)	536 (20.6)	
Body mass index (kg/m^2^)	26.7 ± 4.9	26.4 ± 4.8	0.071
BMI categories (%)			0.140
Normal	305 (40.8)	1083 (41.9)	
Overweight	281 (37.5)	1025 (39.6)	
Obese	162 (21.7)	477 (18.5)	
Diabetes (%)	49 (6.6)	309 (11.7)	<0.001
Number of drinks per week	4 [1–8]	3 [0–8]	<0.001 §
Alcohol consumption (%)	591 (79.0)	1670 (69.9)	<0.001
CESD score	8 [3–13]	8 [4–14]	<0.216 §
Depressed (%)	85 (12.8)	220 (11.6)	0.395

Results are expressed as number of participants (column percentage) for categorical variables and as average ± standard deviation or median [interquartile range] for continuous variables. In the excluded column, numbers might not add to the total due to missing data. Between-group comparisons performed using chi-square for categorical variables and student’s *t*-test or Kruskal-Wallis test (§) for continuous variables.

**Table 2 nutrients-14-05167-t002:** Within-day, time-lagged associations of EMA hunger intensity and eating at time T − 1 predicting psychological states at time T, CoLaus|PsyCoLaus study, Lausanne, Switzerland, 2015–2017.

Variables	Model 1	*p*-Value	Model 2	*p*-Value	Model 3	*p*-Value	Model 4	*p*-Value
Hunger intensity								
Sadness	0.018 (0.008; 0.027)	<0.001	0.003 (−0.008; 0.014)	0.607	0.004 (−0.007; 0.015)	0.447	0.009 (−0.002; 0.020)	0.103
Anxiety	0.027 (0.016; 0.038)	<0.001	0.007 (−0.005; 0.019)	0.259	0.011 (−0.001; 0.023)	0.071	0.017 (0.005; 0.030)	0.006
Feeling active	0.082 (0.061; 0.103)	<0.001	0.029 (0.007; 0.051)	0.009	0.029 (0.007; 0.051)	0.010	0.041 (0.018; 0.063)	<0.001
Feeling lively	0.042 (0.028; 0.056)	<0.001	0.021 (0.007; 0.036)	0.005	0.019 (0.004; 0.034)	0.011	0.021 (0.006; 0.036)	0.006
Feeling distracted	−0.030 (−0.045; −0.015)	<0.001	0.000 (−0.016; 0.016)	0.985	−0.001 (−0.017; 0.015)	0.949	−0.005 (−0.022; 0.011)	0.528
Anhedonia	0.007 (−0.005; 0.020)	0.232	−0.008 (−0.021; 0.006)	0.254	−0.005 (−0.018; 0.009)	0.488	−0.001 (−0.015; 0.013)	0.908
Anger	0.009 (−0.001; 0.020)	0.082	−0.001 (−0.012; 0.011)	0.891	0.001 (−0.010; 0.013)	0.829	0.004 (−0.008; 0.016)	0.509
Slow thinking	−0.036 (−0.049; −0.024)	<0.001	−0.016 (−0.029; −0.003)	0.016	−0.016 (−0.029; −0.003)	0.018	−0.019 (−0.033; −0.006)	0.004
Restlessness	0.007 (−0.006; 0.019)	0.287	−0.010 (−0.024; 0.003)	0.143	−0.009 (−0.023; 0.005)	0.195	−0.003 (−0.016; 0.011)	0.726
Previous eating								
Sadness	−0.015 (−0.060; 0.029)	0.495	−0.035 (−0.080; 0.009)	0.122	0.002 (−0.043; 0.046)	0.934	−0.015 (−0.060; 0.030)	0.511
Anxiety	−0.028 (−0.078; 0.022)	0.268	−0.059 (−0.110; −0.009)	0.021	−0.035 (−0.085; 0.015)	0.173	−0.056 (−0.107; −0.006)	0.030
Feeling active	0.151 (0.058; 0.243)	0.001	0.068 (−0.023; 0.159)	0.144	0.116 (0.025; 0.208)	0.012	0.079 (−0.014; 0.171)	0.095
Feeling lively	0.070 (0.008; 0.132)	0.026	0.026 (−0.034; 0.087)	0.395	0.020 (−0.040; 0.081)	0.509	0.017 (−0.045; 0.078)	0.597
Feeling distracted	−0.086 (−0.152; −0.019)	0.012	−0.031 (−0.097; 0.035)	0.354	−0.052 (−0.118; 0.014)	0.124	−0.038 (−0.105; 0.029)	0.264
Anhedonia	−0.065 (−0.120; −0.009)	0.022	−0.085 (−0.141; −0.030)	0.003	−0.047 (−0.103; 0.009)	0.098	−0.059 (−0.115; −0.002)	0.041
Anger	−0.012 (−0.059; 0.035)	0.620	−0.027 (−0.075; 0.021)	0.264	−0.016 (−0.064; 0.032)	0.512	−0.024 (−0.073; 0.024)	0.331
Slow thinking	−0.080 (−0.134; −0.026)	0.004	−0.038 (−0.091; 0.016)	0.167	−0.031 (−0.084; 0.022)	0.256	−0.019 (−0.073; 0.035)	0.493
Restlessness	0.012 (−0.045; 0.068)	0.684	−0.018 (−0.075; 0.039)	0.534	−0.005 (−0.062; 0.052)	0.868	−0.021 (−0.078; 0.036)	0.470

Results are expressed as coefficient and (95% confidence intervals). Morning (08:00) records were excluded. Model 1 adjusted for sex (male/female), age (continuous), marital status (married/other), smoking (never/former/current), BMI categories (normal/overweight/obese), diabetes (yes/no), alcohol consumption (yes/no), depressive status (yes/no) and previous eating (yes/no). Model 2 included all confounders of Model 1 plus assessment period (12:00, 16:00 or 20:00). Model 3 included all confounders of Model 2 plus the psychological state assessed at time T − 1. Model 4 included all confounders of Model 3 plus quantity of food consumed.

**Table 3 nutrients-14-05167-t003:** Within-day, time-lagged associations of psychological states at time T − 1 predicting EMA hunger intensity at time T, CoLaus|PsyCoLaus study, Lausanne, Switzerland, 2015–2017.

Variables	Model 1	*p*-Value	Model 2	*p*-Value	Model 3	*p*-Value	Model 4	*p*-Value
Sadness	0.035 (0.006; 0.063)	0.016	0.008 (−0.020; 0.036)	0.583	0.009 (−0.019; 0.037)	0.546	0.002 (−0.025; 0.030)	0.871
Anxiety	0.023 (−0.002; 0.049)	0.070	0.022 (−0.003; 0.047)	0.085	0.023 (−0.002; 0.048)	0.066	0.021 (−0.003; 0.045)	0.089
Feeling active	0.028 (0.013; 0.042)	<0.001	0.050 (0.036; 0.064)	<0.001	0.050 (0.036; 0.064)	<0.001	0.048 (0.034; 0.062)	<0.001
Feeling lively	0.023 (0.001; 0.046)	0.044	0.046 (0.024; 0.068)	<0.001	0.045 (0.023; 0.067)	<0.001	0.023 (0.001; 0.046)	0.044
Feeling distracted	−0.019 (−0.039; 0.001)	0.058	−0.039 (−0.059; −0.019)	<0.001	−0.039 (−0.058; −0.019)	<0.001	−0.039 (−0.058; −0.019)	<0.001
Anhedonia	0.028 (0.005; 0.052)	0.019	0.002 (−0.021; 0.025)	0.849	0.003 (−0.020; 0.027)	0.768	−0.004 (−0.027; 0.018)	0.696
Anger	0.009 (−0.001; 0.020)	0.082	0.008 (−0.018; 0.035)	0.535	0.009 (−0.017; 0.036)	0.490	0.007 (−0.019; 0.032)	0.609
Slow thinking	−0.023 (−0.046; 0.001)	0.063	−0.057 (−0.080; −0.034)	<0.001	−0.057 (−0.080; −0.034)	<0.001	−0.059 (−0.082; −0.036)	<0.001
Restlessness	0.031 (0.008; 0.054)	0.008	0.040 (0.018; 0.062)	<0.001	0.040 (0.018; 0.063)	<0.001	0.038 (0.016; 0.059)	0.001

Results are expressed as coefficient and (95% confidence intervals). Morning (08:00) records were excluded. Model 1 adjusted for sex (male/female), age (continuous), marital status (married/other), smoking (never/former/current), BMI categories (normal/overweight/obese), diabetes (yes/no), alcohol consumption (yes/no), depressive status (yes/no) and previous eating (yes/no). Model 2 included all confounders of Model 1 plus assessment period (12:00, 16:00 or 20:00). Model 3 included all confounders of Model 2 plus the psychological state assessed at time T − 1. Model 4 included all confounders of Model 3 plus quantity of the food consumed.

## Data Availability

The data of CoLaus|PsyCoLaus study used in this article cannot be fully shared as they contain potentially sensitive personal information on participants. According to the Ethics Committee for Research of the Canton of Vaud, sharing these data would be a violation of the Swiss legislation with respect to privacy protection. However, coded individual-level data that do not allow researchers to identify participants are available upon request to researchers who meet the criteria for data sharing of the CoLaus|PsyCoLaus Datacenter (CHUV, Lausanne, Switzerland). Any researcher affiliated to a public or private research institution who complies with the CoLaus|PsyCoLaus standards can submit a research application to research.colaus@chuv.ch or research.psycolaus@chuv.ch. Proposals requiring baseline data only, will be evaluated by the baseline (local) Scientific Committee (SC) of the CoLaus and PsyCoLaus studies. Proposals requiring follow-up data will be evaluated by the follow-up (multicentric) SC of the CoLaus|PsyCoLaus cohort study. Detailed instructions for gaining access to the CoLaus|PsyCoLaus data used in this study are available at www.colaus-psycolaus.ch/professionals/how-to-collaborate/.

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
