# Peer review of "Associations between Hunger and Psychological Outcomes: A Large-Scale Ecological Momentary Assessment Study"

_nutrients, 2022, doi:10.3390/nu14235167_

Round 1
Reviewer 1 Report
I believe that the conclusions should be worked on a little more. It is true that they respond to the objectives, but it would help if the conclusion included an assessment of the importance or significance that the results and discussion of this study can have on patient care and care procedures.
Author Response
Please see the attachment "Reviewer 1.docx"

Reviewer 2 Report
This is a well written manuscript. Please seem my comments below.
Introduction
You do a good job writing about why it's important to understand the interaction between hunger and psychological states. However, I think that you could expand upon the big picture why (i.e. is it because of obesity rates?). I think giving a clearer picture of why it is important to understand this association on a larger scale is very important.
Methodology
I appreciate the authors sharing the link for where we may find more information on the study however, I would suggest that on lines 81-82, you should perhaps cite the seminal study to help the reader.
For lines 140-141 please cite the studies that report validity and reliability of the CES-D with the cut-off score used to estimate presence/absence of depression.
Why did you not control for the size of the meal in your analysis?
Did you check normality of distribution of the continuous variables? Were they all normally distributed? If not, what techniques were used to normalize the variables?
Did you correct for multiple analyses? With 4 models you increase the risk for Type I errors
I understand that you recruited everyone that you could for this study so I think it's difficult to conduct an a priori power analysis. However, did you perform a post-hoc power analysis? It would help strengthen your argument in the discussion section about how you identified small, but significant associations.
Results:
I appreciate the authors presenting their results succintly. I also appreciate the fact that these authors compared the sample that was included to the individuals who did not complete the study/declined to participate.
By 12am you mean mid-night correct? How many individuals responded to the survey at midnight? Or do you mean 12pm (noon)? Based on the way the study is designed I think you might mean 12pm??? Perhaps presenting the times as 0800, 1200, 1600, 2000 might be a better option
Otherwise you do a good job of presenting your results.
Discussion
You do a good job in your discussion of identifying the limitations of your study however, you do not bring up the literature on food timing and/or the interaction between food timing and sleep on psychological states. Significant community based work has been conducted by Drs. Jansen, O'Brien and Hosmer on this. There is evidence from some of Dr. Jansen's work that skipping breakfast resulted in increased feelings of fatigue. I would recommend identifying the potential impact that meal timing and/or sleep could have on psychological states in your limitations sections.
Please expand on your conclusion. Please write a conclusion paragraph when you sum up your study and provide a take home message
Author Response
Please see attachment "Reviewer 2.docx"

Reviewer 3 Report
The Manuscript (nutrients-1954127) title " Associations between hunger and psychological outcomes: a 2 large-scale ecological momentary assessment study" evaluates a very interesting topic on the correlations between hunger and psychological outcomes.
The Manuscript appears well-written and clear to read, but it requires major revisions.
Specific comments:
In the table 1 there were problems in the age groups: 45-55, 55-65, and 65+, but is should be better to indicate 45-55 and 56-65. Age ranges should be revised in this section.
Line 310 The association between previous eating and psychological states should be discussed considering the following references:
Chen et al., 2021. DOI: 10.1111/joss.12715
Sanna et al., 2021. DOI: 10.3389/fnins.2021.599593
Author Response
Please see attachment "Reviewer 3.docx"

Round 2
Reviewer 2 Report
I appreciate the authors addressing my concerns.
Regarding the authors comments on the post-hoc power analysis, I would recommend using G*Power to calculate post-hoc power. It's a free software and easy to use.
I personally have not used Stata in years, but the scikit-learn (sklearn) library in Python has a power calculation function.
I hope these suggestions help the authors.
Another suggestion for this manuscript is to remove your comments
Reviewer 3 Report
I appreciate the revised draft of the Manuscript (nutrients-1954127). However, some minor revisions are required:
Specific comments:
Line 322-324 in the Discussion this sentence is wrong. Probably Authors would indicate that subjects with olfactory and gustatory deficits are more prone to develop depression (reference 30) and obesity (reference 31).
In the Discussion section Line 327-328 Authors should delete the p values from the text.
